# Efficient state-space modularization for planning: theory, behavioral and neural signatures

**Daniel McNamee, Daniel Wolpert, Máté Lengyel**
Computational and Biological Learning Lab
Department of Engineering
University of Cambridge
Cambridge CB2 1PZ, United Kingdom
{d.mcnamee|wolpert|m.lengyel}@eng.cam.ac.uk

## Abstract

Even in state-spaces of modest size, planning is plagued by the "curse of dimensionality". This problem is particularly acute in human and animal cognition given the limited capacity of working memory, and the time pressures under which planning often occurs in the natural environment. Hierarchically organized modular representations have long been suggested to underlie the capacity of biological systems [1,2] to efficiently and flexibly plan in complex environments. However, the principles underlying efficient modularization remain obscure, making it difficult to identify its behavioral and neural signatures. Here, we develop a normative theory of efficient state-space representations which partitions an environment into distinct modules by minimizing the average (information theoretic) description length of planning within the environment, thereby optimally trading off the complexity of planning across and within modules. We show that such optimal representations provide a unifying account for a diverse range of hitherto unrelated phenomena at multiple levels of behavior and neural representation.

## 1 Introduction

In a large and complex environment, such as a city, we often need to be able to flexibly plan so that we can reach a wide variety of goal locations from different start locations. How might this problem be solved efficiently? Model-free decision making strategies [3] would either require relearning a policy, determining which actions (e.g. turn right or left) should be chosen in which state (e.g. locations in the city), each time a new start or goal location is given – a very inefficient use of experience resulting in prohibitively slow learning (but see Ref. 4). Alternatively, the state-space representation used for determining the policy can be augmented with extra dimensions representing the current goal, such that effectively multiple policies can be maintained [5], or a large "look-up table" of action sequences connecting any pair of start and goal locations can be represented – again leading to inefficient use of experience and potentially excessive representational capacity requirements.

In contrast, model-based decision-making strategies rely on the ability to simulate future trajectories in the state space and use this in order to flexibly plan in a goal-dependent manner. While such strategies are data- and (long term) memory-efficient, they are computationally expensive, especially in state-spaces for which the corresponding decision tree has a large branching factor and depth [6]. Endowing state-space representations with a hierarchical structure is an attractive approach to reducing the computational cost of model-based planning [7-11] and has long been suggested to be a cornerstone of human cognition [1]. Indeed, recent experiments in human decision-making have gleaned evidence for the use and flexible combination of "decision fragments" [12] while neuroimaging work has identified hierarchical action-value reinforcement learning in humans [13] and indicated that

dorsolateral prefrontal cortex is involved in the passive clustering of sequentially presented stimuli when transition probabilities obey a "community" structure [14].

Despite such a strong theoretical rationale and empirical evidence for the existence of hierarchical state-space representations, the computational principles underpinning their formation and utilization remain obscure. In particular, previous approaches proposed algorithms in which the optimal state-space decomposition was computed based on the optimal solution in the original (non-hierarchical) representation [15,16]. Thus, the resulting state-space partition was designed for a specific (optimal) environment solution rather than the dynamics of the planning algorithm itself, and also required *a priori* knowledge of the optimal solution to the planning problem (which may be difficult to obtain in general and renders the resulting hierarchy obsolete). Here, we compute a hierarchical modularization optimized for planning directly from the transition structure of the environment, without assuming any *a priori* knowledge of optimal behavior. Our approach is based on minimizing the average information theoretic description length of planning trajectories in an environment, thus explicitly optimizing representations for minimal working memory requirements. The resulting representation are hierarchically modular, such that planning can first operate at a global level across modules acquiring a high-level "rough picture" of the trajectory to the goal and, subsequently, locally within each module to "fill in the details".

The structure of the paper is as follows. We first describe the mathematical framework for optimizing modular state-space representations (Section 2), and also develop an efficient coding-based approach to neural representations of modularised state spaces (Section 2.6). We then test some of the key predictions of the theory in human behavioral and neural data (Section 3), and also describe how this framework can explain several temporal and representational characteristics of "task-bracketing" and motor chunking in rodent electrophysiology (Section 4). We end by discussing future extensions and applications of the theory (Section 5).

## 2 Theory

### 2.1 Basic definitions

In order to focus on situations which require flexible policy development based on dynamic goal requirements, we primarily consider discrete "multiple-goal" Markov decision processes (MDPs). Such an MDP, $\mathbb{M} := \{\mathcal{S}, \mathcal{A}, \mathcal{T}, \mathcal{G}\}$, is composed of a set of states $\mathcal{S}$, a set of actions $\mathcal{A}$ (a subset $A_s$ of which is associated with each state $s \in \mathcal{S}$), and transition function $\mathcal{T}$ which determines the probability of transitioning to state $s_j$ upon executing action $a$ in state $s_i$, $p(s_j|s_i, a) := \mathcal{T}(s_i, a, s_j)$. A *task* $(s, g)$ is defined by a *start state* $s \in \mathcal{S}$ and a *goal state* $g \in \mathcal{G}$ and the agent's objective is to identify a *trajectory* of *via states* $\mathbf{v}$ which gets the agent from $s$ to $g$. We define a *modularization*[1] $\mathcal{M}$ of the state-space $\mathcal{S}$ to be a set of Boolean matrices $\mathcal{M} := \{M_i\}_{i=1...m}$ indicating the module membership of all states $s \in \mathcal{S}$. That is, for all $s \in \mathcal{S}$, there exists $i \in 1, \ldots, m$ such that $M_i(s) = 1$, $M_j(s) = 0 \ \forall j \neq i$. We assume this to form a disjoint cover of the state-space (overlapping modular architectures will be explored in future work). We will abuse notation by using the expression $s \in M$ to indicate that a state $s$ is a member of a module $M$. As our planning algorithm $\mathcal{P}$, we consider random search as a worst-case scenario although, in principle, our approach applies to any algorithm such as dynamic programming or Q-learning[3] and we expect the optimal modularization to depend on the specific algorithm utilized.

We describe and analyze planning as a Markov process. For planning, the underlying state-space is the same as that of the MDP and the transition matrix $T$ is a marginalization over a planning policy $\pi_{\text{plan}}$ (which, here, we assume is the random policy $\pi_{\text{rand}}(a|s_i) := \frac{1}{|A_{s_i}|}$)

$$T_{ij} = \sum_a \pi_{\text{plan}}(a|s_i)\, \mathcal{T}(s_i, a, s_j) \tag{1}$$

Given a modularization $\mathcal{M}$, planning at the global level is a Markov process $M_{\text{G}}$ corresponding to a "low-resolution" representation of planning in the underlying MDP where each state corresponds

to a "local" module $M_i$ and the transition structure $T_\text{G}$ is induced from $T$ via marginalization and normalization[22] over the internal states of the local modules $M_i$.

## 2.2 Description length of planning

We use an information-theoretic framework[23,24] to define a measure, the (expected) *description length* (DL) of planning, which can be used to quantify the complexity of planning $\mathcal{P}$ in the induced global $L(\mathcal{P}|M_\text{G})$ and local modules $L(\mathcal{P}|M_i)$. We will compute the DL of planning, $L(\mathcal{P})$, in a non-modularized setting and outline the extension to modularized planning DL $L(\mathcal{P}|\mathcal{M})$ (elaborating further in the supplementary material). Given a task $(s,g)$ in an MDP, a solution $\mathbf{v}^{(n)}$ to this task is an $n$-state trajectory such that $\mathbf{v}_1^{(n)} = s$ and $\mathbf{v}_n^{(n)} = g$. The description length (DL) of this trajectory is $L(\mathbf{v}^{(n)}) := -\log p_\text{plan}(\mathbf{v}^{(n)})$. A task may admit many solutions corresponding to different trajectories over the state-space thus we define the DL of the task $(s,g)$ to be the expectation over all trajectories which solve this task, namely

$$L(s,g) := \mathbb{E}_{\mathbf{v},n}\left[L(\mathbf{v}^{(n)})\right] = -\sum_{n=1}^{\infty}\sum_{\mathbf{v}^{(n)}} p(\mathbf{v}^{(n)}|s,g)\, \log p(\mathbf{v}^{(n)}|s,g) \tag{2}$$

This is the $(s,g)$-th entry of the *trajectory entropy* matrix $\mathbb{H}$ of $\mathbb{M}$. Remarkably, this can be expressed in closed form[25]:

$$[\mathbb{H}]_{sg} = \sum_{v \neq g}[(I - T_g)^{-1}]_{sv}\, H_v \tag{3}$$

where $T$ is the transition matrix of the planning Markov chain (Eq. 1), $T_g$ is a sub-matrix corresponding to the elimination of the $g$-th column and row, and $H_v$ is the *local entropy* $H_v := H(T_{v\cdot})$ at state $v$. Finally, we define the description length $L(\mathcal{P})$ of the planning process $\mathcal{P}$ itself over all tasks $(s,g)$

$$L(\mathcal{P}) := \mathbb{E}_{s,g}[L(s,g)] = \sum_{(s,g)} P_s\, P_g\, L(s,g) \tag{4}$$

where $P_s$ and $P_g$ are priors of the start and goal states respectively which we assume to be factorizable $P_{(s,g)} = P_s\, P_g$ for clarity of exposition. In matrix notation, this can be expressed as $L(\mathcal{P}) = P_s\, \mathbb{H}\, P_g^\mathsf{T}$ where $P_s$ is a row-vector of start state probabilities and $P_g$ is a row-vector of goal state probabilities.

The planning DL, $L(\mathcal{P}|\mathcal{M})$, of a nontrivial modularization of an MDP requires (1) the computation of the DL of the global $L(\mathcal{P}|M_\text{G})$ and the local planning processes $L(\mathcal{P}|M_i)$ for global $M_\text{G}$ and local $M_i$ modular structures respectively, and (2) the weighting of these quantities by the correct priors. See supplementary material for further details.

## 2.3 Minimum modularized description length of planning

Based on a modularization, planning can be first performed at the global level across modules, and then subsequently locally within the subset of modules identified by the global planning process (Fig. 1). Given a task $(s,g)$ where $s$ represents the *start state* and $g$ represents the *goal state*, global search would involve finding a trajectory in $M_\text{G}$ from the induced initial module (the unique $M_s$ such that $M_s(s) = 1$) to the goal module ($M_g(g) = 1$). The result of this search will be a *global directive* across modules $M_s \to \cdots \to M_g$. Subsequently, local planning sub-tasks are solved within each module in order to "fill in the details". For each module transition $M_i \to M_j$ in $M_\text{G}$, a local search in $M_i$ is accomplished by planning from an entrance state from the previous module, and planning until an exit state for module $M_j$ is entered. This algorithm is illustrated in Figure 1.

By minimizing the sum of the global $L(\mathcal{P}|M_\text{G})$ and local DLs $L(\mathcal{P}|M_i)$, we establish the optimal modularization $\mathcal{M}^*$ of a state-space for planning:

$$\mathcal{M}^* := \arg\min_{\mathcal{M}} \left[L(\mathcal{P}|\mathcal{M}) + L(\mathcal{M})\right], \text{ where } L(\mathcal{P}|\mathcal{M}) := L(\mathcal{P}|M_\text{G}) + \sum_i L(\mathcal{P}|M_i) \tag{5}$$

Note that this formulation explicitly trades-off the complexity (measured as DL) of planning at the global level, $L(\mathcal{P}|M_\text{G})$, i.e. across modules, and at the local level, $L(\mathcal{P}|M_i)$, i.e. within individual modules (Fig. 1C-D). In principle, the representational cost of the modularization itself $L(\mathcal{M})$ is also

part of the trade-off, but we do not consider it further here for two reasons. First, in the state-spaces considered in this paper, it is dwarfed by the the complexities of planning, $L(\mathcal{M}) \ll L(\mathcal{P}|\mathcal{M})$ (see the supplementary material for the mathematical characterization of $L(\mathcal{M})$). Second, it taxes long-term rather than short-term memory, which is at a premium when planning[26,27]. Importantly, although computing the DL of a modularization seems to pose significant computational challenges by requiring the enumeration of a large number of potential trajectories in the environment (across or within modules), in the supplementary material we show that it can be computed in a relatively straightforward manner (the only nontrivial operation being a matrix inversion) using the theory of finite Markov chains[22].

## 2.4   Planning compression

The planning DL $L(s, g)$ for a specific task $(s, g)$ describes the expected difficulty in finding an intervening trajectory $\mathbf{v}$ for a task $(s, g)$. For example, in a binary coding scheme where we assign binary sequences to each state, the expected length of string of random 0s and 1s corresponding to a trajectory will be shorter in a modularized compared to a non-modularized representation. Thus, we can examine the relative benefit of an optimal modularization, in the Shannon limit, by computing the ratio of trajectory description lengths in modularized and non-modularized representations of a task or environment[28]. In line with spatial cognition terminology[29], we refer to this ratio as the *compression factor* of the trajectory.

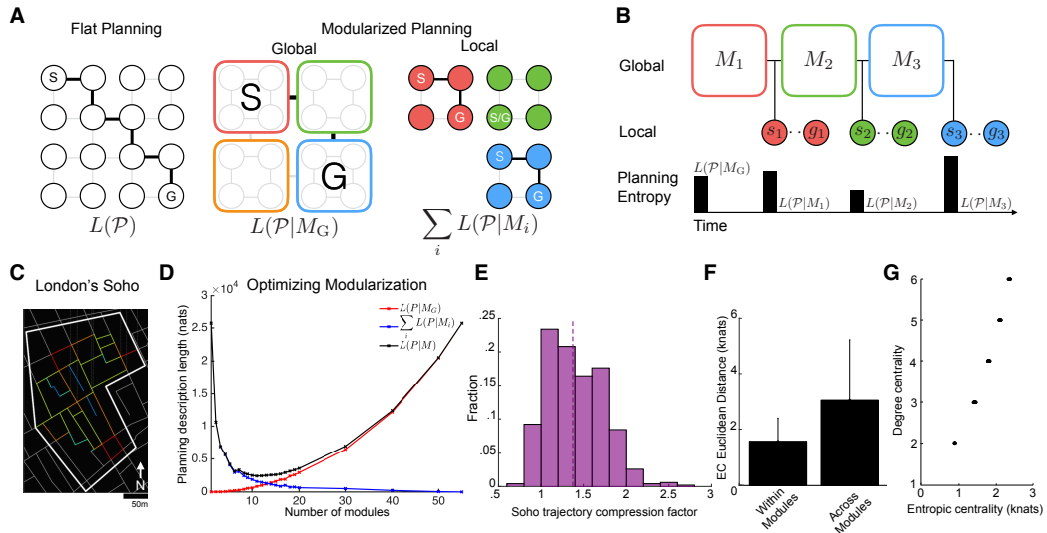

**Figure 1. Modularized planning. A.** Schematic exhibiting how planning, which could be highly complex using a flat state space representation (left), can be reformulated into a hierarchical planning process via a modularization (center and right). Boxes (circles or squares) show states, lines are transitions (gray: potential transitions, black: transitions considered in current plan). Once the "global directive" has been established by searching in a low-resolution representation of the environment (center), the agent can then proceed to "fill in the details" by solving a series of local planning sub-tasks (right). Formulae along the bottom show the DL of the corresponding planning processes. **B.** Given a modularization, a serial hierarchical planning process unfolds in time beginning with a global search task followed by local sub-tasks. As each global/local planning task is initiated in series, there is a phasic increase in processing which scales with planning difficulty in the upcoming module as quantified by the local DL, $L(\mathcal{P}|M_i)$. **C.** Map of London's Soho state-space, streets (lines, with colors coding degree centrality) correspond to states (courtesy of Hugo Spiers). **D.** Minimum expected planning DL of London's Soho as a function of the number of modules (minimizing over all modularizations with the given number of modules). Red: global, blue: local, black: total DL. **E.** Histogram of compression factors of 200 simulated trajectories from randomly chosen start to goal locations in London's Soho. **F.** Absolute entropic centrality (EC) differences within and across connected modules in the optimal modularization of the Soho state-space. **G.** Scatter plot of degree and entropic centralities of all states in the Soho state-space.

## 2.5 Entropic centrality

The computation of the planning DL (Section 2.2) makes use of the *trajectory entropy* matrix $\mathbb{H}$ of a Markov chain. Since $\mathbb{H}$ is composed of weighted sums of local entropies $H_v$, it suggests that we can express the contribution of a particular state $v$ to the planning DL by summing its terms for all tasks $(s, g)$. Thus, we define the *entropic centrality*, $E_v$, of a state $v$ via

$$E_v = \sum_{s,g} D_{svg} H_v \tag{6}$$

where we have made use of the *fundamental tensor of a Markov chain D* with components $D_{svg} = \left[ (I - T_g)^{-1} \right]_{sv}$. Note that task priors can easily be incorporated into this definition. The entropic centrality (EC) of a state measures its importance to tasks across the domain and its gradient can serve as a measure of "subgoalness" for the planning process $\mathcal{P}$. Indeed, we observed in simulations that one strategy used by an optimal modularization to minimize planning complexity is to "isolate" planning DL within rather than across modules, such that EC changes more across than within modules (Fig. 1F). This suggests that changes in EC serve as a good heuristic for identifying modules.

Furthermore, EC is tightly related to the graph-theoretic notion of *degree centrality* (DC). When transitions are undirected and are deterministically related to action, degree centrality $\deg(v)$ corresponds to the number of states which are accessible from a state $v$. In such circumstances and assuming a random policy, we have

$$E_v = \sum_{s,g} D_{svg} \frac{1}{\deg(v)} \log(\deg(v)) \tag{7}$$

The ECs and DCs of all states in a state-space reflecting the topology of London's Soho are plotted in Fig. 1G and show a strong correlation in agreement with this analysis. In Section 3.2 we test whether this tight relationship, together with the intuition developed above about changes in EC demarcating approximate module boundaries, provides a normative account of recently observed correlations between DC and human hippocampal activity during spatial navigation[30].

## 2.6 Efficient coding in modularized state-spaces

In addition to "compressing" the planning process, modularization also enables a neural channel to transmit information (for example, a desired state sequence) in a more efficient pattern of activity using a hierarchical entropy coding strategy[31] whereby contextual codewords signaling the entrance to and exit from a module constrain the set of states that can be transmitted to those within a module thus allowing them to be encoded with shorter description lengths according to their relative probabilities[28] (i.e. a state that forms part of many trajectory will have a shorter description length than one that does not). Assuming that neurons take advantage of these strategies in an efficient code[32], several predictions can be made with regard to the representational characteristics of neuronal populations encoding components of optimally modularized state-spaces. We suggest that the phasic neural responses (known as "start" and "stop" signals) which have been observed to encase learned behavioral sequences in a wide range of control paradigms across multiple species[33–36] serve this purpose in modularized control architectures. Our theory makes several predictions regarding the temporal dynamics and population characteristics of these start/stop codes. First, it determines a specific temporal pattern of phasic start/stop activity as an animal navigates using an optimally modularized representation of a state-space. Second, neural representations for the start signals should depend on the distribution of modules, while the stop codes should be sensitive to the distribution of components within a module. Considering the minimum average description length of each of these distribution, we can make predictions regarding how much neural resources (for example, the number of neurons) should be assigned to represent each of these start/stop variables. We verify these predictions in published neural data[36,34] in Section 4.

## 3 Route compression and state-space segmentation in spatial cognition

### 3.1 Route compression

We compared the compression afforded by optimal modularization to a recent behavioral study examining trajectory compression during mental navigation[29]. In this task, students at the University

of Toronto were asked to mentally navigate between a variety of start and goal locations on their campus and the authors computed the (inverse) ratio between the duration of this mental navigation and the typical time it would physically take to walk the same distance. Although mental navigation time was substantially smaller than physical time, it was not simply a constant fraction of it, but instead the ratio of the two (the compression factor) became higher with longer route length (Fig. 2A). In fact, while in the original study only a linear relationship between compression factor and physical route length was considered, reanalysing the data yielded a better fit by a logarithmic function ($R^2 = 0.69$ vs. $0.46$).

In order to compare our theory with these data, we computed compression factors between the optimally modularized and the non-modularized version of an environment. This was because students were likely to have developed a good knowledge of the campus' spatial structure, and so we assumed they used an approximately optimal modularization for mental navigation, while the physical walking time could not make use of this modularization and was bound to the original non-modularized topology of the campus. As we did not have access to precise geographical data about the part of the U. Toronto campus that was used in the original experiment, we ran our algorithm on a part of London Soho which had been used in previous studies of human navigation[30]. Based on 200 simulated trajectories over route lengths of 1 to 10 states, we found that our compression factor showed a similar dependence on route length[2] (Fig. 2B) and again was better fit by a logarithmic versus a linear function ($R^2 = 0.82$ vs. $0.72$, respectively).

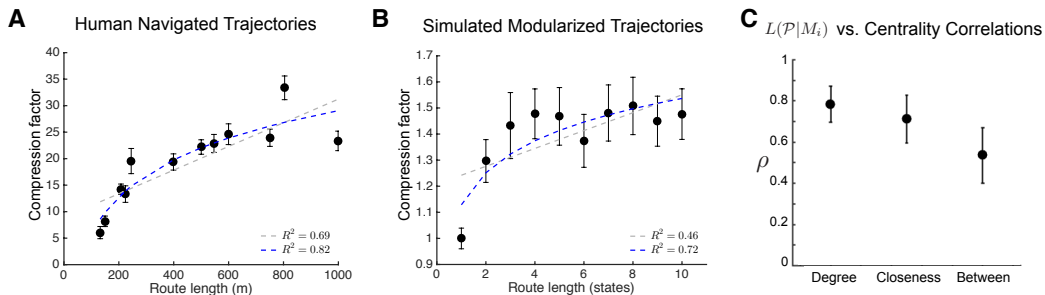

**Figure 2. Modularized representations for spatial cognition. A.** Compression factor as a function of route length for navigating the U. Toronto campus (reproduced from Ref. 29) with linear (grey) and logarithmic fits (blue). **B.** Compression factors for the optimal modularization in the London Soho environment. **C.** Spearman correlations between changes in local planning DL, $L(\mathcal{P}|M_i)$, and changes in different graph-theoretic measures of centrality.

### 3.2 Local planning entropy and degree centrality

We also modeled a task in which participants, who were trained to be familiar with the environment, navigated between randomly chosen locations in a virtual reality representation of London's Soho by pressing keys to move through the scenes[30]. Functional magnetic resonance imaging during this task showed that hippocampal activity during such self-planned (but not guided) navigation correlated most strongly with changes in a topological state "connectedness" measure known as *degree centrality* (DC, compared to other standard graph-theoretic measures of centrality such as "betweenness" and "closeness"). Although changes in DC are not directly relevant to our theory, we can show that they serve as a good proxy for a fundamental quantity in the theory, planning DL (see Eq. 7), which in turn should be reflected in neural activations.

To relate the optimal modularization, the most direct prediction of our theory, to neural signals, we made the following assumptions (see also Fig. 1B). 1. Planning (and associated neural activity) occurs upon entering a new module (as once a plan is prepared, movement across the module can be automatic without the need for further planning, until transitioning to a new module). 2. The magnitude of neural activity is related to the local planning DL, $L(\mathcal{P}|M_i)$, of the module (as the higher the entropy, the more trajectories need to be considered, likely activating more neurons with different tunings for state transitions, or state-action combinations[37], resulting in higher overall

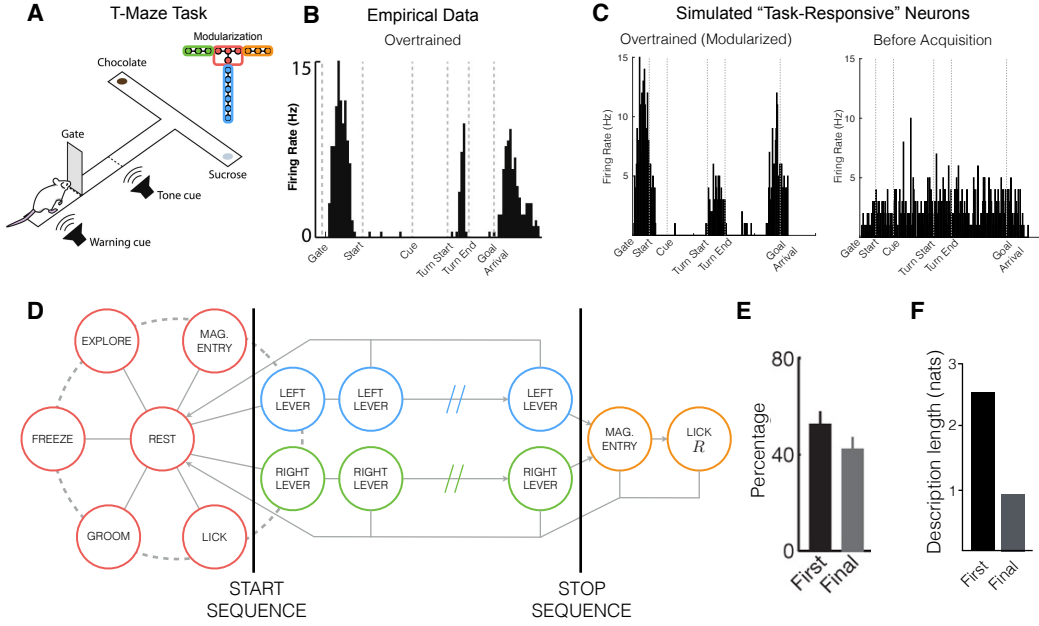

**Figure 3. Neural activities encoding module boundaries. A.** T-maze task in which tone determines the location of the reward (reproduced from Ref. 34). Inset: the model's optimal modularization of the discretized T-maze state-space. Note that the critical junction has been extracted to form its own module which isolates the local planning DL caused by the split in the path. **B.** Empirical data exhibiting the temporal pattern of task-bracketing in dorsolateral striatal (DLS) neurons. Prior to learning the task, ensemble activity was highly variable both spatially and temporally throughout the behavioral trajectory. Reproduced from Ref. 34. **C.** Simulated firing rates of "task-responsive" neurons after and before acquiring an optimal modularization. **D.** The optimal modularization (colored states are in the same module) of a proposed state-space for an operant conditioning task[36]. Note that the lever pressing sequences form their own modules and thus require specialized start/stop codes. **E.** Analyses of striatal neurons suggesting that a larger percentage of neurons encoded lever sequence initiations compared to terminations, and that very few encoded both. Reproduced from Ref. 36. **F.** Description lengths of start/stop codes in the optimal modularization.

activity in the population). Furthermore, as before, we also assume that participants were sufficiently familiar with Soho that they used the optimal modularization (as they were specifically trained in the experiment). Having established that under the optimal modularization entropic centrality (EC) tends to change more across than within modules (Fig. 1F), and also that EC is closely related to DC (Fig. 1G), the theory predicts that neural activity should be timed to changes in DC. Furthermore, the DLs of successive modules along a trajectory will in general be positively correlated with the differences between their DLs (due to the unavoidable "regression to the mean" effect[3]). Noting that the planning DL of a module is just the (weighted) average EC of its states (see Section 2.5), the theory thus more specifically predicts a positive correlation between neural activity (representing the DLs of modules) and changes in EC and therefore changes in DC – just as seen in experiments.

We verified these predictions numerically by quantifying the correlation of changes in each centrality measure used in the experiments with transient changes in local planning complexity as computed in the model (Fig. 2C). Across simulated trajectories, we found that changes in DC had a strong correlation with changes in local planning entropy (mean $\rho_{\text{deg}} = 0.79$) that was significantly higher ($p < 10^{-5}$, paired t-tests) than the correlation with the other centrality measures. We predict that even higher correlations with neural activity could be achieved if planning DL according to the optimal modularization, rather than DC, was used directly as a regressor in general linear models of the fMRI data.

# 4    Task-bracketing and start/stop signals in striatal circuits

Several studies have examined sequential action selection paradigms and identified specialized task-bracketing[33,34] and "start" and "stop" neurons that are invariant to a wide range of motivational, kinematic, and environmental variables[36,35]. Here, we show that task-bracketing and start/stop signals arise naturally from our model framework in two well-studied tasks, one involving their temporal[34] and the other their representational characteristics[36].

In the first study, as rodents learned to navigate a T-maze (Fig. 3A), neural activity in dorsolateral striatum and infralimbic cortex became increasingly crystallized into temporal patterns known as "task-brackets"[34]. For example, although neural activity was highly variable before learning; after learning the same neurons phasically fired at the start of a behavioral sequence, as the rodent turned into and out of the critical junction, and finally at the final goal position where reward was obtained. Based on the optimal modularization for the T-maze state-space (Fig. 3A inset), we examined spike trains from a simulated neurons whose firing rates scaled with local planning entropy (see supplementary material) and this showed that initially (i.e. without modularization, Fig. 3C right) the firing rate did not reflect any task-bracketing but following training (i.e. optimal modularization, Fig. 3C left) the activity exhibited clear task-bracketing driven by the initiation or completion of a local planning process. These result show a good qualitative match to the empirical data (Fig. 3B, from Ref. 34) showing that task-bracketing patterns of activity can be explained as the result of module start/stop signaling and planning according to an optimal modular decomposition of the environment.

In the second study, rodents engaged in an operant conditioning paradigm in which a sequence of eight presses on a left or right lever led to the delivery of high or low rewards[36]. After learning, recordings from nigrostriatal circuits showed that some neurons encoded the initiation, and fewer appeared to encode the termination, of these action sequences. We used our framework to compute the optimal modularization based on an approximation to the task state-space (Fig. 3D) in which the rodent could be in many natural behavioral states (red circles) prior to the start of the task. Our model found that the lever action sequences were extracted into two separate modules (blue and green circles). Given a modularization, a hierarchical entropy coding strategy uses distinct neural codewords for the initiation and termination of each module (Section 2.6). Importantly, we found that the description lengths of start codes was longer than that of stop codes (Fig. 3F). Thus, an efficient allocation of neural resources predicts more neurons encoding start than stop signals, as seen in the empirical data (Fig. 3E). Intuitively, more bits are required to encode starts than stops in this state-space due to the relatively high level of entropic centrality of the "rest" state (where many different behaviors may be initiated, red circles) compared to the final lever press state (which is only accessible from the previous Lever press state and where the rodent can only choose to enter the magazine or return to "rest"). These results show that the start and stop codes and their representational characteristics arise naturally from an efficient representation of the optimally modularized state space.

# 5    Discussion

We have developed the first framework in which it is possible to derive state-space modularizations that are directly optimized for the efficiency of decision making strategies and do not require prior knowledge of the optimal policy before computing the modularization. Furthermore, we have identified experimental hallmarks of the resulting modularizations, thereby unifying a range of seemingly disparate results from behavioral and neurophysiological studies within a common, principled framework. An interesting future direction would be to study how modularized policy production may be realized in neural circuits. In such cases, once a representation has been established, neural dynamics at each level of the hierarchy may be used to move along a state-space trajectory via a sequence of attractors with neural adaptation preventing backflow[38], or by using fundamentally non-normal dynamics around a single attractor state[39]. The description length that lies at the heart of the modularization we derived was based on a specific planning algorithm, random search, which may not lead to the modularization that would be optimal for other, more powerful and realistic, planning algorithms. Nevertheless, in principle, our approach is general in that it can take any planning algorithm as the component that generates description lengths, including hybrid algorithms that combine model-based and model-free techniques that likely underlie animal and human decision making[40].

## Footnotes

[1]This is an example of a "propositional representation" [17,18] and is analogous to state aggregation or "clustering" [19,20] in reinforcement learning which is typically accomplished via heuristic bottleneck discovery algorithms [21]. Our method is novel in that it does not require the optimal policy as an input and is founded on a normative principle.

[2]Note that the absolute scale of our compression factor is different from that found in the experiment because we did not account for the trivial compression that comes from the simple fact that it is just generally faster to move mentally than physically.

[3]Transitioning to a module with larger/smaller DL will cause, on average, a more positive/negative DL change compared to the previous module DL.

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
