[Supplementary Material · nips2016CR_supp.pdf]

# Supplementary Material for NIPS 2016 Paper #2245
# Efficient state-space modularization for planning: theory, behavioral and neural signatures

## Contents

## 1 Modularized description length of planning

The global planning DL, $L(\mathcal{P}|M_{\mathrm{G}})$, can be easily computed after marginalizing over the internal states of each module. Defining $P_{\mathrm{S}}$ ($P_{\mathrm{G}}$) to be the prior over start (goal) modules $P_{\mathrm{S}}(M_i) := \sum_{s \in M_i} P_s$ $\left(P_{\mathrm{G}}(M_i) := \sum_{g \in M_i} P_g\right)$, then $L(\mathcal{P}|M_{\mathrm{G}}) = P_{\mathrm{S}} \, \mathbb{H}_{\mathrm{G}} \, P_{\mathrm{G}}^{\mathsf{T}}$ where $\mathbb{H}_{\mathrm{G}}$ is the trajectory entropy across modules. In order to compute the local planning DL: $L(\mathcal{P}|M_i) := (P_{\mathrm{S}}(M_i) + P_{\mathrm{V}}(M_i) + P_{\mathrm{G}}(M_i)) \sum_{s_i, g_i \in M_i} P_{s_i} P_{g_i} L(s_i, g_i|M_i)$ we must first establish the induced priors over "sub-tasks" $(s_i, g_i)$ within the module $M_i$. The probability that a state $s_i \in M_i$ serves as a sub-start state is the probability that $s_i$ is the entrance state to $M_i$ given a module transition into $M_i$ at the global level. Conversely, a state $g_i \in M_i$ is a sub-goal state if it is the last transient state within $M_i$ before a trajectory transitions out of the module $M_i$. These probabilities, as well as

**Figure 4.** Two state-space modularizations exhibiting the relationship between state-space structure and modularization compression are shown. Following modularization, the first state-space is broken up into four modules corresponding to the two large "room" as well as a split in each room. Even in an a relatively homogeneous state-space (such as each of these rooms) planning complexity is minimized if the space is partitioned into equally sized modules. In the second example, the "corridor" is extracted exhibiting the partitioning of the state-space into modules with relatively constant entropic centralities. In the final column, we simulate 200 random walks and state-goal trajectories in each state-space and compute compression factors (2.4). Two types of state sequences are considered, namely a *random walk* is a random sequence of states while a *state-goal trajectory* is a sequence of states generated by a planning policy. The set of trajectories is a subset of the set of walks since the goal state cannot be repeatedly visited in a trajectory. The first example shows that the first MDP is relatively incompressible while the second exemplifies the fact that minimizing modularized DL specifically compresses solutions to problems $(s, g)$ (i.e. trajectories) in the environment.

the probability $P_V(M_i)$ that $M_i$ is transiently accessed under the global directive, can be computed precisely based on the fundamental matrix of a Markov chain (see Section 2). A special case, which is computed separately, is when the start and goal states are within the same module. This contributes no additional DL at the global level but is added as a separate cost in the local planning entropy calculations.

In order to compute the optimal modularization, we currently use a brute-force algorithm, which takes around <10 mins to modularize the Soho state-space (55 states, 134 transitions). In future work, we aim to incorporate more sophisticated optimization techniques such as parallelization, greedy submodular optimization and genetic algorithms. Code is available online at https://github.com/dmcnamee.

## 2   Module transition probabilities

We constructed (see Section 2.5, main text) the fundamental tensor $D$ of the global planning process

$$[D]_{SVG} := (I - T_G)^{-1}_{SV} \tag{1}$$

where $S$ indicates a start module, $V$ a transient module, and $G$ a goal module. We record a useful property of the fundamental tensor. The probability that a module $V$ is transiently accessed given a goal module $G$ and a start module $S$ is[1]

$$\left[ D_{SVG} \times \left( \mathrm{diag}_{SV} \, D_{SVG}^{-1} \right) \right]_{SV} \tag{2}$$

This expression, weighted by the prior probabilities of tasks $(S, G)$ gives the prior probability of a module being accessed over all tasks

$$P_V(M_V) = \sum_S [P_S D_{SVG} \times \left(\text{diag}_{SV} D_{SVG}^{-1}\right) P_G^T]_{SV} \tag{3}$$

We obtain the transient module transition probabilities $P(M_i \rightarrow M_j)$ by considering the global goal-module absorbing chain with fundamental matrix $N_G = (I - T_G)^{-1}$ and summing over all global tasks $(S, G)$ weighted by the task priors $P_S$ and $P_G$:

$$P(M_i \rightarrow M_j) \quad = \quad P(M_j | M_i) P(M_i) = \left[P_S D_{SVG} \times \left(\text{diag}_{SV} D_{SVG}^{-1}\right) P_G^T \times T|_{G^\perp, G}\right]_{ij} \tag{4}$$

## 2.1 Across-modules transitions and sub-start probabilities

Let us consider two connected modules, $M_i$ and $M_j$, in our modularization $\mathcal{M}$ and consider the probability that an entrance state $s_j \in S_j$ is accessed from start state $s_i \in S_i$. It is known from the theory of finite Markov chains (Theorem 3.5.4 in Ref. 1) that

$$P(s_{in} = s_j, M_j | s_{in} = s_i, M_i) = \left[N_{M_i} \times T|_{M_i, M_j}\right]_{ij} \tag{5}$$

where $T|_{M_i, M_j}$ denotes the restriction of $T$ to the row-components corresponding to the states of $M_i$ and the column-components of $M_j$. Summing over the states of $s_i \in M_i$ gives the probability $P(s_{in} = s_j, M_j | M_i)$ that $s_j \in M_j$ is an sub-start state given that the global directive has identified a transition from $M_j$.

## 2.2 Within-modules transitions and sub-goal probabilities

We assume that we have an entrance state $s_a$ and an exit state $s_b$ in a module $M_i$. The probability of $s_b$ being an exit state from the module is the probability that it is transiently accessed before exiting to module $M_j$ (Theorem 3.5.7 in Ref. 1)

$$P(s_{out} = s_j | s_{in} = s_i, M_i) = \left[N_{M_i} \times \text{diag}(N_{M_i})^{-1} \times T|_{M_i, M_j}\right]_{ij} \tag{6}$$

# 3 Alternate formulation of trajectory entropy

We use the formulation for trajectory entropy in a Markov chain established in Ref. 2. This refines and extends a previous expression derived in Ref. 3 which we record here:

$$\mathbb{H} := K - \hat{K} + \bar{s}^{-1} L(\mathbf{v}_\infty)_\Delta \tag{7}$$

where we have

$$
\begin{aligned}
K &:= \frac{H^* - L(\mathbf{v}_\infty)}{I - T + A^x} \\
\hat{K}_{ij} &= K_{jj} \ , \ \forall j \\
H^*_{ij} &:= H(T_{i\cdot}) \\
L(\mathbf{v}_\infty)_{ii} &= \bar{s}_i^{-1} H(T) \\
L(\mathbf{v}_\infty)_{ij} &= 0 \ , \ \forall i \neq j \\
A_{ij} &= \bar{s}_j
\end{aligned}
\tag{8}
$$

We verified numerically that these two different formulations matched in a wide range of Markov chains.

# 4 Representational cost of modularization

We quantify the cost of representing a modularization via the expected description length of randomly producing a particular modularization[4,5]. Such as process has two components, namely the specification of the number of modules $n_M$ (which must be between 1 and the cardinality of the state-space

$|\mathcal{S}|$) and the assignment of each state to a module.

$$
\begin{aligned}
L(\mathcal{M}) &= -\log(P(\mathcal{M})) \\
&= -\log(P(\mathcal{M}|n_M)) - \log(P(n_M)) \\
&= -\log\left[\prod_{s\in\mathcal{S}} P(s \in M | n_M)\right] - \log(P(n_M)) \\
&= \sum_{s\in\mathcal{S}} \log(n_M) + \log(|\mathcal{S}|) \\
&= |S|\log(n_M) + \log(|\mathcal{S}|) \qquad\qquad (9)
\end{aligned}
$$

# 5 Experimental datasets and analysis details

## 5.1 Spatial navigation task

Functional magnetic resonance imaging (fMRI) was used to study the brain activity of human subjects engaged in spatial navigation in London's Soho (Fig. 1C, main text). All subjects were students of University College London and therefore tended to be highly familiar with the environment. In addition, subjects were evaluated to ensure that they had prior knowledge of the environment, after completing a training process in which (1) they studied maps and photographs of the state-space locations, (2) they were given a guided tour of the area, and (3) practised the task that they would perform in the scanner[6].

On each trial, after first orienting the subjects at the start state and identifying the goal state, subjects watched first-person-view movies of travel along novel start-goal trajectories through Soho. Half of the trials required subjects to make decisions as to how to best proceed in order to complete the task. Specifically, prior to arriving at a junction in the state-space, participants indicated with a button press which subsequent direction to travel in. In control trials, subjects were instructed to press a button indicating a particular direction of travel rather than choosing themselves.

The fMRI data was analyzed with general linear models containing regressors corresponding to time series of centrality measures (betweenness, closeness, and degree) and changes thereof. The key result (see Section 3.2, main text), is that hippocampal activity was specifically sensitive to changes in degree centrality (as opposed to closeness or betweenness). Further details can be found in the publication[6] of this study.

## 5.2 Task-bracketing simulations

We simulated the spiking activity of Poisson neurons whose firing rate was driven by the initialization and termination of modules, and local planning entropy (within modules), in a non-modularized, and an optimally modularized version, of the T-maze state-space (Fig. 3A inset) used in Ref. 7. We assumed a baseline firing rate of 5KHz, a refractory period of 10ms, and a neural gain of 20 relating the encoded variables (start/stop, planning) to the firing rate in order to match the range of empirically observed firing rates[7]. After generating a trajectory, we resampled the time course of task variable signals to match the sampling frequency of 1000KHz. Fig. 3C (main text, modularized on the left) shows the perievent time histogram of 10 simulated trials of the ensemble activity. The median firing rate was equalized across the two conditions. We assumed that, on arrival at the goal, rodents shifted to a new behavioral module corresponding to the consummation of the reward which transiently increased planning entropy in addition to a module stop signal. Without this, there is still a clear peak at the goal arrival timepoint but with a lower average firing rate.

## 5.3 Operant conditioning state-space

We designed a model state-space of the operant conditioning paradigm used in Ref. 8 incorporating the fixed-ratio reward schedule relating sequences of 8 lever presses to reward delivery. In addition to the behavioral states ("lever press", "magazine entry", "lick") directly related to the action-outcome contingencies, rodents in the chamber may engage in a range of additional behaviors thus we included a range of alternative behaviors in the state-space model, namely "grooming", "resting", "freezing", and "exploring". All states connected by the dashed lines are directly accessible from one another.

For example, we assume the rodent can be in a "lick" state directly following an "explore" state without transitioning through "rest". The efficient modular decomposition displayed in Fig. 3D (main text) does not strongly depend on the structure of the state-space adjacent to the "rest" state and is mainly dependent on the natural nonuniform task distribution whereby the only rewarding goal state is "licking when reward is present" and the rodent is initialized in the "rest" state. The plotted description lengths correspond to the initialization and termination of the "lever" action sequences (as modules at the "global" level) under the stationary transition distribution.

## 6 Comparison with other measures of planning complexity/difficulty

Planning description length $L(\mathcal{P}|\mathcal{M})$ is a scalar measure which allows MDPs to be ranked in terms of the complexity of finding, or encoding, a solution based on a planning process $\mathcal{P}$ given a modularization $\mathcal{M}$. Note that this is distinct from the formal computational complexity theory of MDPs as a problem domain which classes them as $P$-complete[9]. In a set of 17 small MDPs, designed to span a variety of state-space topologies and task priors, we compared planning DL against a variety of alternative planning complexity and difficulty measures, namely (1) the expected shortest path length[10], (2) the expected path length (generated by the planning process), (3) the number of states in the MDP, (4) the number of transitions in the MDP, and (5) the average degree centrality. See Fig. 5 for scatter plots for the first four measures (see Fig. 1G, main text for a plot of entropic centrality versus degree centrality). Of these, we found that expected path length ($R^2 = 0.69$), the number of transitions ($R^2 = 0.46$), and the average degree centrality ($R^2 = 0.62$), significantly explained variability in PDL in a linear model ($p < 0.05$).

Although expected path length is significantly correlated with planning description length in our set of MDPs, it is easy to generate counter-examples to this effect. Consider an MDP consisting solely of deterministic "forward" transitions along a "corridor" of states from a start state at one end to a goal at the other (i.e. without actual choices). Here, DL agrees with intuition, assigning minimal complexity, independent of corridor length, while expected path length assigns a larger complexity, increasing with corridor length. This is exemplified by the data point in Fig. 5 with the lowest planning DL (DL equals 0, expected path length equals 6). Therefore, one can expect that state-space modularizations based on expected path length will "spend" modules on breaking up deterministic state sequences where no planning is required. Mathematically, the critical difference is the multiplication by local entropy in the planning DL measure. This sets to zero the contribution of transient states which do not contribute to overall trajectory entropy.

## 7 Comparing efficient modularizations and optimal behavioral hierarchies

The "optimal behavioral hierarchy"[11] (OBH) approach seeks to find the state-space decomposition which "best explains" the optimal trajectories. This objective is formalized as a bayesian model selection over the possible state-space hierarchies:

$$P(behavior|hierarchy) \propto \sum_{\pi \in \Pi} P(behavior|hierarchy, \pi)P(\pi|hierarchy) \qquad (10)$$

where $\Pi$ is the set of all behavioral policies which can be generated from a particular hierarchy.

This approach is distinct from that of efficient modularization (EM). First, OBH requires the optimal policy to be known before it can be applied. If used with a planning policy (such as random search) instead, as we do, it does not result in a meaningful modularization. The modularization would depend on the intrinsic stochasticity of planning via the generation of the *behavior* variable. Even if one were to optimize Eq. (10) based on the average or minimal paths of an ensemble of planning *behaviors*, such an optimized hierarchy would compress the description of such planned trajectories well but not necessarily compress the generation of them.

Second, the objectives are fundamentally different in that even if one was to use the optimal policy with EM, the modularization can be quite different from that drawn from an OBH (for examples, see Fig. 6). This is because we directly optimize for the memory requirements (see main text) whereas OBH-optimized representations would still require large capacity for maintaining the "meta-actions" of the optimal policy (in long-term memory), and for storing the resulting trajectories (in working memory). To illustrate this numerically, we established the optimal trajectories $\pi_{\text{opt}}$

**Figure 5.** We computed planning description length for a variety of deterministic MDPs. For each plot, MDPs are gray-scaled in order of increasing planning DL. Significant linear relationships are indicated by a least-squares line. Planning description length is measured in nats. We describe the planning difficulty measure along the x-axes in each panel. Note that, for expected shortest path length and expected path length, the expectation of the corresponding variables under the task distribution $P(s, g)$ was used. **Expected shortest path length.** The optimal trajectories for every task $(s, g)$ was computed and the number of states in each trajectory counted (including the goal state). **Expected path length.** The expected number of steps until arrival at the goal state under a random search planning process was computed. **Number of states.** The number of states in the MDP (independent of task prior). **Number of transitions.** The number of transitions in the MDP (independent of task prior).

(i.e. minimal paths) for all $16 \times 15 = 240$ tasks $(s, g)$ in MDP 2 (Fig. 6), and computed the total description lengths for each trajectory: (1) $L_{EM}$, based on the partition defined by efficient modularization, and (2) $L_{OBH}$, based on the partition computed via optimal behavioral hierarchy. For all trajectories, the total description length based on EM was smaller with the average difference being $\frac{1}{240} \sum (L_{OBH}(\pi_{\text{opt}}) - L_{EM}(\pi_{\text{opt}})) = 181.69$nats. A similar analysis of trajectories generated by a random policy $\pi_{\text{rand}}$ led to the same conclusion with an average difference of $\frac{1}{240} \sum (L_{OBH}(\pi_{\text{rand}}) - L_{EM}(\pi_{\text{rand}})) = 1901.33$nats. In a behavioral experiment, one could test whether the distributions of compression factors exhibited by subjects while planning in a calibrated set of MDPs and task distributions, were better fit by EM or OBH partitions.

## 8 Entropic centrality and state-space bottlenecks

Strongly modular decision-making environments tend to have "bottleneck" states at the interfaces between modules. From a graph-theoretic point of view, these are states which bridge between clusters of highly connected states. For planning, they serve as important "waypoints" since many trajectories must necessarily travel through them[12]. Bottlenecks are often the focus of "subgoal" discovery algorithms, based on which, temporally extended action sequences or "options" may be defined[13]. Behavioral experiments have shown[11] that human subjects can identify such bottleneck states despite only having experienced local state-state transitions and never observed the global, "bird's eye" view of the entire state-space as displayed in Fig. 7A,C.

**Figure 6.** Maximally efficient modularizations and optimal behavioral hierarchies [11] are presented for two distinct MDPs designed to highlight differences in the corresponding partitions. For both MDPs, we assume the agent may be required to navigate between any two states with equal probability. Partitions are color-coded. **MDP 1.** This homogeneous "open space" is decomposed in the efficient modularization framework but does not contain an optimal behavioral hierarchy. The log model evidence $\log P(behavior|\text{OBH})$ (Eq. 10) in favor of the OBH hierarchy compared to the EM hierarchy is $\log P(behavior|\text{OBH}) - \log P(behavior|\text{EM}) = 168.25$. **MDP 2.** The transition structure has been altered to reflect a more modular structure (the number of states remains the same). EM extracts the "corridor" as a distinct module however the OBH has only two modules with some redundancy (the color-gradient states may be assigned, together, to either module). In this case, $\log P(behavior|\text{OBH}) - \log P(behavior|\text{EM}) = 33.08$.

It appears that efficient modularization tends to partition the environment based on changes in the entropic centrality of states (see main text). Here, we examine whether the magnitude of the entropic centrality gradient across the state-space can serve as a measure of state "bottleneckness" in an MDP using a discrete analogue of the Laplacian operator [14]. In Fig. 7A,C, we exhibit the state-space graphs of two MDPs previously used for human behavioral experiments of state bottleneck identification [11]. One can observe, from a global viewpoint, that both of these state-spaces consists of two "rooms" linked by a "corridor". Note that, in Fig. 7A, all states have the same degree centrality of three (the number of states connected to a given state). Despite this, subjects successfully [11] identified the corridor states as bottlenecks in a "bus-stop placement" task (see Ref. 11 for descriptions of the behavioral experiments).

We compute the magnitude $|\nabla E_v|$ of the entropic centrality gradient $\nabla E_v$ at state $v$ analogously to the discrete "umbrella"[1] Laplacian operator [14] $\Delta$ based on the relation $\nabla^2 = \Delta$:

$$|\nabla E_v| = \sqrt{\sum_{n \in \mathcal{N}_1(v)} T_{nv}(E_n - E_v)^2} \qquad (11)$$

where $\mathcal{N}_1(v)$ is the neighbourhood of states which are directly connected, via nonzero transitions, to state $v$ and $T$ is the planning transition structure of the environment (Eq. 1, main text). After computing entropic centrality gradient magnitudes at each state for each task $(s, g)$, $|\nabla E_v|$ is the expectation of this random vector over the task prior $P(s, g)$.

In Fig. 7B,D, we scale the node sizes of the environments in Fig. 7A,C according to $|\nabla E_v|$ based on a uniform distribution of tasks $(s, g)$ revealing how $|\nabla E_v|$ captures the degree to which a state is a

**Figure 7. A.** MDP used for behavioral experiments in Solway et al. (see Fig. 2C there). Importantly, this MDP has been designed such that each of the ten states has the same degree centrality (three) despite the fact that there is a "bottleneck" between the upper and lower "rooms". **B.** Node sizes are scaled according to entropic centrality gradient magnitude (Eq. 11) of the corresponding state showing that these scalar values serve as a measure of state "bottleneckness". The two state with the highest entropic centrality gradient magnitude are highlighted in grey. Interestingly, our measure $\Delta E$ also assigns the second highest value to the states which seemed to be the second most consistent choice of subjects when probed to identify bottleneck states. **C.** MDP used for behavioral experiments in Solway et al. (see Fig. 2D there). Note the clear bottleneck state between the "rooms". **D.** Node sizes are scaled according to the gradient magnitude of entropic centrality (Eq. 11). The scale is reduced compared to B in order to account for the larger number of states which globally increases entropic centrality. The state with the highest entropic centrality gradient magnitude is highlighted in grey. Our measure assigns the highest value to the bottleneck state and the second highest value to the states of high connectivity positioned at the center of the two rooms.

bottleneck in the global planning structure of the environment. This measure could aid in the discovery of subgoals, especially given that it does not require the pre-computation of the optimal policy as with previous methods[15]. Furthermore, behavioral experiments could be performed in order to test whether the apparent sensitivity of humans to state-space bottlenecks is reflective of a wider cognitive state-space representation strategy based on gradients in entropic centrality. Potentially one could implicitly infer such cognitive representations from compression factor distributions since explicitly probing subjects to reveal their perceived bottlenecks may be confounded by other considerations. For example, subjects may reasonably place a "bus-stop" specifically at the center of a long directed corridor in order to minimize the expected path length of state-goal trajectories even though this does not correspond to a bottleneck state and does not alter the complexity of planning.

## Footnotes

[1]This particular discrete approximation to the Laplacian operator is appropriate for our situation since the state-space has little geometric structure. If, for example, these state-spaces were embedded in a Riemannian manifold, this would induce a measure of the angle between states. Variants of Eq. 11, which incorporate more geometric structure, could be used in such a scenario.