[Reviews · NeurIPS 2016]

Reviewer 1

Summary

The authors look at the problem of finding a state-space modularization in path planning problems formalized as MDPs. The idea is to use the MDL criterion on the modularization and the plans given the modularization. By splitting the planning into planning at the global level across modules and the local level within modules, one can trade off the complexity of planning on global and local levels given a modularization. The modularization is formalized as dividing states in the original MDP into non-overlapping modules. Planning is then carried out on the modules and within the modules themselves. The planning process considered here is random search, which allows the transitions from one plan to the next to be expressed as a Markov process. Utilizing a previous result (ref. 25) allows efficiently computing the DL of planning L(P|M). The authors apply their developed theory to derive measures of degree of modularization and compression factor for route length and suggest to use these to explain results from a number of behavioral and neuronal data.

Qualitative Assessment

The paper feels like a Current Biology submission reformatted for Nips. There is a lot of material very densely arranged in the manuscript, which makes it difficult for me to grasp the computational details at the level that would have been desirable for a Nips submission. Sections 2.2 and 2.5 are not trivial, and the details that have been put in the supplementary material seem much more relevant compared to all the experimental results that the authors want to account for. So, this looks like three papers in one manuscript: the theory, applications of the theory to behavioral data, and applications to neuronal data. I was not able to follow the details regarding entropic centrality and the relationship to degree centrality. Some details are missing about the simulations carried out as mentioned in 4.1. The assumptions made for relating planning DL to neuronal activations need more explanation for my taste. There is also relevant other work that the authors may consider citing, e.g. Gershman, S. J., Pesaran, B., & Daw, N. D. (2009) that finds evidence for factored representations and Rothkopf, C. A., & Ballard, D. H. (2010), which uses a modularized MDP formulation for modeling local navigation. I very much like the manuscript and the science therein and I can see it have significant impact on the understanding of hierarchical representations for planning, I just would have liked it to be more focused on the theoretical material for a Nips submission.

Confidence in this Review

1-Less confident (might not have understood significant parts)


Reviewer 2

Summary

This paper provides a new algorithm for sub-dividing a state space for planning and search. The algorithm is shown to provide qualitative fits to features of human navigation data and neural data from rodents in a simple learning task.

Qualitative Assessment

The paper is very ambitious and develops a computational model of how the state space can be carved up (aggregated?) for planning. This model is applied to some intriguing data on human and rodent spatial navigation and seems to nicely pull together disparate threads from the literature. Unfortunately, the exposition was sufficiently abstract, so that following the thread from the model to the results (simulations) was challenging, leaving unclear exactly how the model was explaining the behaviour and making evaluation difficult. I unpack these and a couple of other issues below. 1. It is not entirely clear how the different ideas introduced in the paper (modularity, centrality, description length) fit together into a single model of behaviour and the brain. From the text, it was not clear to me how the simulations and predictions for the different behavioural tasks were generated. I think the account in the text is just too abstract with insufficient detail of how to get from those three equations to a full model of the behaviour (which they obviously have). 2. Almost no detail is given on the brain simulations (Fig 3C). The results look great, but I have no idea how they were generated from the model as spelled out in the early sections. These first two issues may be addressed in the supplemental materials (which I did not read), but my understanding is that the core paper needs to stand alone, with the supplementals supplementing the main text, not replacing key aspects of the results. 3. It was not clear to me how their concept of modularization was different than simple state aggregation. That is, the novelty and distinctiveness of the modularization approach was not apparent, and this parallel is even briefly acknowledged in footnote 1, yet the approach is still deemed novel. The paper could still be interesting even if the modularization approach just builds on or even just implements state aggregation with interesting connections to human and animal behaviour, but that was not done here. 4. The optimal modularization presented in Figure 4D seems like it might depend strongly on the “states” that are included in the model. Why are explore, freeze, rest, and groom only behaviours permitted near the start sequence? Unlike some of the experiments with humans on navigation, the behavioral repertoire of the animals is not strongly limited by their location. Animals can engage in these behaviours at any time (though they might not in practice). This needs further justification. Also, the states there are almost all actions, not really states.

Confidence in this Review

2-Confident (read it all; understood it all reasonably well)


Reviewer 3

Summary

This paper presents a theoretical framework for measuring planning complexity in terms of information-theoretic description length. The framework is leveraged to find an information-theoretically optimal modularization of the state-space that reduces total description length. The authors then show that this leads to modules consistent with both behavioral and neural data.

Qualitative Assessment

Major comments: - On the whole I thought this paper was quite interesting and well-written. The authors do a good job showing the explanatory value of their theoretical framework. - It was not entirely clear to me why description length is an appropriate measure of planning difficulty. Why is this better or worse than other measures (e.g., average planning time). - I think there needs to be a better discussion of relation between this work and the work of Solway & Botvinick. Is the only difference algorithmic, or are the underlying objective functions different? Also, the Solway paper reports a number of behavioral experiments which seem to be relevant to the issues at hand. - p. 6: I didn't understand what regression to mean effect meant in this context. Minor comments: - p. 2: "is a set" -> "as a set" - p. 5: "many trajectory" -> "many trajectories"

Confidence in this Review

3-Expert (read the paper in detail, know the area, quite certain of my opinion)


Reviewer 4

Summary

The authors presented a theoretical framework for hierarchical state-space modularization for planning. The paper introduces the problem and the information-theoretic framework to describe, solve and evaluate planning over and within the hierarchical modules. The authors demonstrate that their approach allows a simplified planning across the modules, minimizing the average information description length of trajectories. Thereafter, a subsequent within-module planning can fill the details of the trajectory. Interestingly, their theoretical results matched some neurophysiological/behavioural findings, such as the logarithmic fit of the compression factors between mental navigation and physical time, or the start/stop signals found in rodents during planning tasks. In the former, the compression factor can be found analytically as the ratio between the description lengths in modularized and non-modularized representations of the task. In the latter, the magnitude of neural activity is related to local planning of the module, through a measure of connectedness known as degree centrality (which as been previously shown to be correlated with hippocampus activity).

Qualitative Assessment

My personal concern about the paper is about the validation of the theoretical framework with experimental data. Regarding the route compression experiment, the assumption that mental navigation makes use of optimal modularization is not fully convincing. I would prefer to see evidence showing that the optimal modularization found by this method actually matches the mental representation of the state-space. In principle, this evidence could be seen in the correlation between neural activity and start/stop signals in the T-maze task; however, due to the simplicity of the task, the proposed bracketing could be associated with other input features. Finally, in supplementary materials Figure 4 has not been described in the text.

Confidence in this Review

2-Confident (read it all; understood it all reasonably well)


Reviewer 5

Summary

The authors consider the problem of clustering a state space for the purpose of lower-complexity planning/routing. Modularization has clear benefits in this respect: in theory, one first plans a route across clusters, and then the individual routes within each cluster. In practice, how should one go about performing the clustering operation? The authors propose an MDL based approach, wherein a policy $\pi(action|state)$ is assumed to give rise to a transition matrix $T$, which in turn induces a distribution over all paths across modules, and within modules, for any given clustering. The modularization that results in the lowest entropy is chosen as the winner. The authors test the correlation of predictions made according to this modularizations against several datasets, including both neural activity in rodents and route visualization by humans.

Qualitative Assessment

My fundamental problem with this paper is that the authors did not make the operational connection between minimum description length / trajectory entropy and ``complexity'' clear for me. Encoding a path sampled from a low-entropy T will be cheaper in bits than one sampled from a high-entropy T. It is unclear which problem this solves: the authors mention the computational burden of simulating future trajectories, and they later make reference to minimizing memory requirements. The former only makes sense if one (for some reason) needs to simulate trajectories by flipping coins, and the latter only makes sense if one seeks to record or communicate trajectories (which the authors do point out a relevant use case for in passing in Sec. 3). As such, the theory that is presented does not feel justified or relevant to me. Additionally, (not being familiar with the literature) I fail to see the separation between the author's approach and that of others, described starting on Line 44. The authors claim that their method is not reliant on a solution in the original state space, but this appears to be incorrect: the use of the policy \pi (or a policy informed by Qlearning or DP) directly informs the transition matrix T, which directly informs the modularization.

Confidence in this Review

1-Less confident (might not have understood significant parts)


Reviewer 6

Summary

This paper proposes a new method for planning in which the planning state-space may be learned in a modular way without knowledge of an optimal policy based on hierarchical MDL. Significant biological and statistical evidence is provided as to the motivation and design of a modular state-space learning algorithm.

Qualitative Assessment

This is a well written article, which provides significant evidence as to the motivation and analogues for planning modularization. This is important work which is necessary to help build state space partitions/modules to allow us to scale learning tasks to larger overall state spaces. It would be nice if authors provided a more clear view of the algorithm itself and its implementation (although the supplementary material is very helpful here) - the London Soho path planning dataset is interesting, it would be nice to have a bit more of a clear picture of the dataset and its representation. Some discussion of the computational complexity of the algorithm and the partitioning would be interesting to compare as the partitioning allows us to work on smaller partitioned problems, but some measure for the cost of partitioning would be interesting -- what is the complexity on the example data set? Will there be a reference implementation published so others can reproduce these results?

Confidence in this Review

2-Confident (read it all; understood it all reasonably well)